# Antimicrobial Activity of Protein Fraction from *Naja ashei* Venom against *Staphylococcus epidermidis*

**DOI:** 10.3390/molecules25020293

**Published:** 2020-01-10

**Authors:** Aleksandra Bocian, Ewa Ciszkowicz, Konrad K. Hus, Justyna Buczkowicz, Katarzyna Lecka-Szlachta, Monika Pietrowska, Vladimír Petrilla, Monika Petrillova, Ľubomír Legáth, Jaroslav Legáth

**Affiliations:** 1Faculty of Chemistry, Rzeszow University of Technology, 35-959 Rzeszów, Poland; eciszkow@prz.edu.pl (E.C.); knr.hus@gmail.com (K.K.H.); czaporj@prz.edu.pl (J.B.); szlachta@prz.edu.pl (K.L.-S.); jlegath@prz.edu.pl (J.L.); 2Maria Sklodowska-Curie Institute-Oncology Center, Gliwice Branch, 44-100 Gliwice, Poland; monika.pietrowska@io.gliwice.pl; 3Department of Anatomy, Histology and Physiology, University of Veterinary Medicine and Pharmacy, Komenského 73, 041 81 Kosice, Slovakia; petrillav@gmail.com; 4Zoological Department, Zoological Garden Košice, Široká 31, 040 06 Košice-Kavečany, Slovakia; 5Department of General Education Subjects, University of Veterinary Medicine and Pharmacy, Komenského 73, 041 81 Kosice, Slovakia; monika.petrillova@uvlf.sk; 6Department of Occupational Medicine and Clinical Toxicology, Pavol Jozef Šafárik University Faculty of Medicine and Louis Pasteur University Hospital, Rastislavova 43, 041 90 Košice, Slovakia; lubomir.legath@upjs.sk; 7Department of Pharmacology and Toxicology, University of Veterinary Medicine and Pharmacy, Komenského 73, 041 81 Kosice, Slovakia

**Keywords:** *Naja ashei*, venom proteins, antimicrobial properties, MIC, biofilm

## Abstract

One of the key problems of modern infectious disease medicine is the growing number of drug-resistant and multi-drug-resistant bacterial strains. For this reason, many studies are devoted to the search for highly active antimicrobial substances that could be used in therapy against bacterial infections. As it turns out, snake venoms are a rich source of proteins that exert a strong antibacterial effect, and therefore they have become an interesting research material. We analyzed *Naja ashei* venom for such antibacterial properties, and we found that a specific composition of proteins can act to eliminate individual bacterial cells, as well as the entire biofilm of *Staphylococcus epidermidis*. In general, we used ion exchange chromatography (IEX) to obtain 10 protein fractions with different levels of complexity, which were then tested against certified and clinical strains of *S. epidermidis*. One of the fractions (F2) showed exceptional antimicrobial effects both alone and in combination with antibiotics. The protein composition of the obtained fractions was determined using mass spectrometry techniques, indicating a high proportion of phospholipases A_2_, three-finger toxins, and L-amino acids oxidases in F2 fraction, which are most likely responsible for the unique properties of this fraction. Moreover, we were able to identify a new group of low abundant proteins containing the Ig-like domain that have not been previously described in snake venoms.

## 1. Introduction

*Staphylococcus epidermidis*, a gram-positive, coagulase-negative *Staphylococci* (CNS), is a representative of natural human microbiota inhabiting the skin and mucous membranes. It is estimated that a healthy human organism can be simultaneously inhabited by up to 20 strains of this bacteria without any harm [1]. However, these bacteria can be very dangerous for immunocompromised people and newborns [2]. It causes both primary infections, such as bacteremias, and much more frequent infections associated with various types of medical devices (e.g., catheters, surgical vascular grafts, joint prostheses, heart valves) [3,4]. However, unlike *Staphylococcus aureus*, *S. epidermidis* does not produce toxins and its virulence is determined by its ability to form biofilms that enable them to colonize different types of biomaterials. This biofilm is resistant to antibiotics and prevents the immune response of the host organism [5] due to the presence of the exopolysaccharide matrix [6,7]. Therefore, the treatment of infections caused by biofilm is mainly limited to the replacement of infected medical devices, which significantly increases the cost of therapy [5]. The ability to create an antibiotic-resistant biofilm forced the need for preventive action, focusing mainly on preventive antibiotic therapy in surgical patients. Unfortunately, this strategy, which was demonstrated especially for vancomycin therapy, turned out to be disastrous and led to the emergence of vancomycin-resistant strains [8]. Also, numerous methicillin-resistant *Staphylococcus Epidermidis* (MRSE) [9,10,11], as well as those that are resistant to other antibiotics, including rifamycin, fluoroquinolones, gentamycin, tetracycline, chloramphenicol, erythromycin, clindamycin, and sulphonamides, were described [3].

Therefore, the development of new effective bactericidal agents with a different mechanism of action is an extremely important and urgent problem to be solved [12,13,14]. It is now believed that one of the sources of new compounds with pharmacological potential may be snake venom, which exhibits a wide range of biological activities and may be used in the development of new drugs [15]. It has been known for a long time that both the whole venom of many snake species and its isolated components, e.g., phospholipases A_2_ (PLA_2_s), L-amino acid oxidases (LAAOs), myotoxins, and even their fragments, have antibacterial properties [13,14]. Also, several venom peptides, such as cathelicidin, are bactericidal by inhibiting ATP synthase [15,16]. Therefore, one of the established trends in venomics is the use of “omics” techniques to search for new molecules with antibacterial properties in hitherto undiscovered and rare snake species [17]. To meet these suggestions, we decided to look for proteins with antibacterial properties in the venom of an African spitting cobra, namely *Naja ashei*. It is a relatively poorly described species, which was classified as a separate taxon only in 2007 [18]. In our earlier studies using proteomic techniques, we found that *N. ashei* venom contains, among others, phospholipases A_2_ and 3FTx toxins [19], which have been described many times as having antibacterial properties [20,21,22,23]. Therefore, we decided to fractionate the venom of *Naja ashei* and to investigate the antibacterial activity of individual fractions against *S. epidermidis*.

## 2. Results

### 2.1. Naja ashei Venom Fractionation

To separate proteins from crude *Naja ashei* venom, we performed IEX chromatography on the Resource S column. As the result, 10 fractions were obtained (Figure 1).

The percentage share of individual fractions in the collected material was estimated from the obtained chromatograms using the area under the curve (AUC). The largest part of proteins from the whole pool were found in fractions 4 and 8, the least in fractions 1–2 and 9–10 (Figure 2).

The SDS-PAGE technique was used to monitor the complexity of the fractions. Fractions 6–7 and 1 and 8 were separated into two and three bands, respectively. Fractions 2 and 10 consisted of more than three bands. In fractions 3, 4, 5, and 9, only one band is visible. However, in these samples, as well as in 6 and 8, the lowest bands migrated with the front of the electrophoresis, which means that there is a high probability that there is a mixture of low molecular weight proteins (Figure 3). 

### 2.2. Identification of Proteins in Obtained Fractions

#### 2.2.1. General Characteristics of the Obtained Fractions

MS analysis indicated that in six of the obtained fractions, the predominant group of proteins was 3FTx (F4–F7, F9–F10). In three fractions (F2–F3, F8), the highest share of phospholipases A_2_ was observed, but in F2 it was below 50%. The largest number of different protein groups was found in fraction F1. In fractions F1 and F7, a significant share of SVMPs (snake venom metalloproteinases) was observed while in fractions F5 and F6, the highest percentage of VNGF (venom nerve growth factor) was detected (Figure 4). Detailed information on the identification of proteins in individual fractions can be found in the Appendix A.

In addition, a number of other low abundant proteins were detected. Table 1 shows those with a share exceeding 1% in at least one fraction.

#### 2.2.2. Detailed Composition of the F2 Fraction

Of all the obtained fractions, only F2 showed antibacterial properties (see the next paragraph for details). The protein concentration in this fraction was estimated at 1.49 μg/μL. The main six components of F2 fraction are: PLA_2_s, 3FTxs, CRISPs, LAAOs, alkaline phosphatase-like proteins and Ig-like domain containing proteins. Among the low abundant proteins, three classes of proteases (SVMP, SVSP, and SVCP), CVF, VNGF, and enzyme inhibitors have the highest share (Figure 5).

#### 2.2.3. Proteins with Ig-Like Domain

Overall, there were 38 identifications of proteins belonging to Ig-like superfamily SSF48726; however, only 23 hits were unique. Seventeen unique proteins were identified on the basis of mRNA sequences obtained from the venom glands of *Micrurus* species (15 identifications) or *Boiga irregularis* (2). The rest of them (six proteins) were identified on the basis of genomic sequences from *Ophiophagus hannah*. Two identified proteins were described as transmembrane proteins in the Uniprot database. In turn, two others were also classified as members of the superfamily SSF54452 belonging to MHC class I family. Most proteins from this group were identified in F2 and F1 (19 and 11, respectively). Details on the identifications are summarized in Appendix A. The percentage share of the Ig-like domain-containing protein class in individual fractions is presented in Appendix A.

### 2.3. Microbiological Tests

#### 2.3.1. Determination of the Minimum Inhibitory Concentration (MIC)

All 10 fractions were tested for antibacterial properties. Only fraction F2 demonstrated antimicrobial activity against *S. epidermidis* ATCC 12228 (MIC = 37.25 µg/mL) and *S. epidermidis* ATCC 35984 (MIC = 9.3 µg/mL) strains. The MICs of 9 out of 10 fractions could not be determined, as these fractions were ineffective against *S. epidermidis* even at a concentration of 500 µg/mL. The minimum inhibitory concentration of fraction F2 was 4.6, 9.3, and 37.25 µg/mL, respectively, for clinical *S. epidermidis* strains 2346, 2452, and 2702. The results for the antibiotic inhibitory concentration, MIC, are shown in Table 2.

#### 2.3.2. Synergy Testing

Selected antibacterial compounds combinations in double-dose response (checkerboard) experiments were used to determine the nature of their interaction. The checkerboard assay was conducted using fraction F2, ampicillin, and tetracycline against two studied standard *S. epidermidis* strains and the results are summarized in Table 3, Figure 6 and Figure 7. The checkerboard assay could not be performed on other three MRCNS *S. epidermidis* strains due to the limited amount of fraction F2.

Fraction F2 alone showed high antibacterial activity, with MICs between 4.6 and 37.25 µg/mL, against different *S. epidermidis* strains. In three out of four combinations with ampicillin (AMP) and tetracycline (TET) against two standard strains, the MICs of the fraction were up to 16 times reduced for the TET/fraction F2 combination against *S. epidermidis* ATCC 12228. It was also observed that in all combinations, the MICs of the antibiotics were also reduced in a concentration-dependent manner for fraction F2. The interaction between ampicillin and fraction F2 against *S. epidermidis* ATCC 12228 showed a neutral character while all other combinations demonstrated a synergistic interaction.

#### 2.3.3. Anti-Biofilm Activity

The effects of fraction F2 on the inhibition of biofilm formation were investigated. The biofilm of certified and clinical *S. epidermidis* strains was grown in the presence of a decreasing concentration of fraction F2. The results revealed that the amount of biofilm formed by various strains of *S. epidermidis* was differently affected upon the exposure of fraction F2 (Figure 8).

Fraction F2 at a concentration of 4.6 µg/mL inhibited the formation of 79.02% to 96.91% of the biofilm compared to the control (biofilm formation of certain certified and clinical strains without an anti-biofilm component). The most effective anti-biofilm activity for the lowest concentration (0.58 µg/mL) was demonstrated by fraction F2 against clinical *S. epidermidis* 2702 strain (76.3%) while the highest inhibition of biofilm formation was exhibited against certified ATCC 35984 *S. epidermidis* strain (98.8%).

## 3. Discussion

The activity of snake venom against *Staphylococcus epidermidis* has not been extensively studied so far. The only reports concern viperids, namely *Calloselasma rhodostoma* and *Bothrops atrox* [24] and *Vipera ammodytes* [25]. Bee venom has also been shown to be effective against skin bacteria, in addition to *S. epidermidis*, *Cutibacterium acnes*, and *Streptococcus pyogenes* [26]. Individual venom components are also effective against skin infections, e.g., snake cathelicidin BG-CATH [27], mucroporin-M1 from scorpion venom [28], or *Lymnaea stagnalis* snail peptides [29].

In our experiments, we proved the effectiveness of only one fraction, F2, both in direct bacteria elimination and inhibition of biofilm production. The five main components of this fraction are phospholipases A_2_, 3FTx proteins, L-amino acid oxidases, Ig-like proteins, and CRISPs. The group of proteins with the Ig-like domain has just been described by us for the first time, and so far, there are no functional studies on this group. Therefore, it is difficult to speculate on the participation of this group in the bactericidal effect. As for the other groups, reports indicate their antibacterial character [14]. What is very interesting is that the composition of F2 fraction is quite similar to F1, although there is no LAAOs but there are SVMPs and, more importantly, cathelicidin-like antimicrobial peptides, which have a wide spectrum of antibacterial activity and high efficiency [15,16]. Despite this, the F1 fraction did not show any antibacterial properties in our tests. We surmise the main reason for the lack of these properties may be the 10 times lower concentration of proteins in comparison to fraction F2. 

Almost half of all proteins in the F2 fraction are phospholipases A_2_. Additionally, it is probably the most widely described group of snake venom proteins in terms of antibacterial properties [20,21,30,31,32]. The mechanism of action of these proteins is based on cell wall damage, pore generation in membranes, and their permeabilization [32]. Also, proteins with the three-finger motif (3FTxs) were described as having an antibacterial effect by interacting with the components of membranes and walls of bacteria and their destabilization [22,23,33]. However, the group that was widely described in terms of its antibacterial properties is LAAOs. Interestingly, this group of proteins was not detected during our previous analysis of the *Naja ashei* venom proteome [19]. This may indicate that the share of this group in the whole venom is negligible, but fractionation caused a significant reduction of the sample complexity, thus allowing for the identification of low abundant proteins. There are three hypotheses concerning the antibacterial mechanism of LAAO. The first one assumes that the oxidized form of cofactor present in these enzymes (FAD or FMN) may interact with amino acids, which in turn may affect the structure and function of nucleic acids, proteins, and membranes [34]. The second mechanism results from the fact that during the reaction carried out by these enzymes, hydrogen peroxide appears as a by-product and causes oxidative damage to the membranes [35] or DNA [36]. There is also a hypothesis that LAAOs can directly oxidize amino acids in proteins, thus causing damage [37]. Regardless of whether only one of these mechanisms takes place or all three, the undeniable fact remains that these proteins are extremely effective against bacteria [38,39,40,41]. In the context of the other results obtained from fractionation, the latter group of proteins seems to be the key component with antibacterial properties. This is due to the fact that it is the only fraction in which the amount of LAAO exceeds 1%. Nevertheless, it cannot be excluded that other components of this fraction, such as PLA_2_ or 3FTx described above, act together to create a synergistic antibacterial effect. It also seems probable that the right proportions of the individual components are also necessary to achieve the desired effect. On the basis of the results obtained, we can speculate that the mechanism of action responsible for the efficiency of the F2 fraction is probably due to damage to the bacterial cell wall and outflow of the cytosol content. However, in order to finally answer the question about which ingredient(s) are responsible for the antibacterial effect, detailed analyses of individual ingredients and their combinations are necessary. 

There are three other interesting aspects of our experiments. Firstly, the F2 fraction shows synergistic effects with antibiotics (ampicillin and tetracycline). Despite the fact, that tetracycline and ampicillin belong to two different classes of antibiotics and use different mechanisms of action against bacteria, for both antibiotics, a similar effect for antimicrobial peptides was previously described [42,43,44] and for the whole venom of *Bothrops moojeni* [45]. The antibacterial effect of ampicillin is based on the inhibition of penicillin-binding proteins involved in cell wall synthesis, whereas tetracycline inhibits 30S subunit during protein synthesis [46]. In general, FIC smaller than MIC makes the protein or peptide factor less toxic and the antibiotic more effective at lower concentrations [44,47]. The mechanism of the creation of synergy is explained in many ways. It is possible, for example, that both substances have the same molecular target but do not compete for a binding site [48]. The second possibility is to block the same metabolic pathway at different stages [49,50]. In other cases, the second component somehow strengthens the action of the first component by facilitating access to the molecular target. This is how antibacterial peptides work, which cause permeabilization of the membrane, which in turn causes the release of antibiotics into the periplasm and cytoplasm of bacterial cells [42,51,52,53,54,55,56]. It is also known that *Staphylococci* have an efflux pump, which determines drug resistance to a large extent because it is responsible for removing from cells various types of substances, which are unnecessary for metabolism or harmful to cells [57]. *Bothrops moojeni* venom inhibits the action of such a pump in *Staphylococcus aureus* and thus, in combination with antibiotics, increases their effectiveness [45]. Perhaps, therefore, one of the components of the F2 fraction may also have similar properties. The synergistic effect of the F2 fraction with ampicillin and tetracycline obtained by us promises very good prospects for the future. As the promising strategies for solving the problem of drug resistance are, on the one hand, the search for new antibacterial factors and, on the other hand, the use of combined antibiotics with other compounds, such as peptides or proteins [58]. Therefore, it seems that this result will be a very important point for future research. 

The second important aspect of the results obtained in this experiment is the ability of the F2 fraction to inhibit biofilm formation. *S. epidermidis* is genetically conditioned to live in the form of biofilm, which is proved by the downregulation of basic cellular processes, such as biosynthesis of nucleic acids, proteins, and wall-building elements [59]. This explains the relatively high resistance to antibiotics, which acts against actively growing cells [60,61]. These bacteria produce poly-γ-glutamic acid (PGA) and poly-*N*-acetylglucosamine (PNAG) exopolymers, which play a key role in the resistance of bacterial cells to human antibacterial peptides and avoidance of phagocytosis by neutrophils [62,63,64]. There are few reports describing the action of venom and its components against biofilm formation. Therefore, the information that the F2 fraction inhibits the biofilm of both certified and clinical drug-resistant strains is extremely important. Until now, it was known that the whole venom of *Bothrops moojeni* inhibits the formation of biofilm [45] as well as the lectin from *Bothrops jaracussu* [65,66]. In all three cases, the inhibition of the biofilm did not result in the destruction of cells or the slowing down of their growth. This is very important because, usually, biofilms are insensitive to antibiotics and the increase in the effectiveness of antibiotics is explained by the ability to inhibit biofilm formation [67,68,69], which could explain why the synergistic effect in the case of F2 fractions was more pronounced for the strain forming the biofilm (ATCC 35984).

The last very interesting discovery was the identification of proteins with the Ig-like domain. According to the latest data concerning *Naja ashei*, its venom is composed mostly of 3FTxs (69%) and PLA_2_s (27%). However, there are also small amounts of SVMPs (2.1%), VNGF (1%), CRISPs (0.7%), CVF (0.12%), and 5′-nucleotidases (0.014%) [19]. After fractionation, we were able to discover other groups of low abundant proteins that were previously described in other snake species, for example, L-amino acid oxidases or snake venom serine proteases (SVSPs) (Table 1). Moreover, we discovered the group of Ig-like domain-containing proteins, which, as far as we know, has not yet been described for snake venom. According to our calculations, this group constitutes less than 1% of all proteins collected during chromatography, i.e., less than 1% of all venom proteins of *Naja ashei*. It seems that this group is extremely low abundant, which could explain why it has not been noticed so far. 

The assignment of the identified proteins to the Ig-like domain-containing protein group was made based on information from the superfamily database of structural and functional annotation [70] on the UniProt website. The assigned superfamily SSF48726 is a superfamily of immunoglobulins, which brings together beta proteins with the immunoglobulin-like beta-sandwich {48725} fold type. According to the integrated resource of protein families, domains, and functional sites, all of the identified proteins have immunoglobulin-like domain (IPR007110) and immunoglobulin-like fold IPR013783 [71]. Unfortunately, the Ig-like domain is one of the most widespread but also the most heterogeneous domains. Proteins with this fold differ in their cellular localization, amino acid sequence, and biological role [72]. Apart from the actual immunoglobulins, proteins possessing this domain also include, for example, enzymes [73], their inhibitors [74], transcription factors [75], or components of ion channels [76]. Moreover, what is very interesting is that some snake venom proteins also have this domain [77,78,79]. However, multiple sequence alignment by Clustal Omega showed that the similarity of these proteins to the identified Ig-like domain-containing proteins is low and does not exceed 30%. Given the variety of functions of proteins in this superfamily, it is difficult to speculate what their role in the venom of *Naja ashei* is, but at this stage of the study, it appears that the identified proteins are unlikely to be constituents of the immune system or blood contaminants, especially considering that they were mostly identified by transcripts from the venom glands [78,80,81] and that there are no reports that the actual immunoglobulins are synthesized there. Given the high heterogeneity of this family, it is very difficult to speculate what the function of these proteins is. Further research is needed to answer a number of questions: Do all the proteins identified in our studies have the same or similar function, do they play a role in envenoming, or do they have a function in the proper functioning of the venom glands? The obtained results suggest that *Naja ashei* venom may contain proteins not previously described in any of the known snake venoms and it cannot be ruled out that these proteins are more common (at least within the *Naja* genus).

## 4. Materials and Methods 

### 4.1. Venom Fractionation

For the experiments, pooled venom of *Naja ashei* from two adult snakes (male and female) was used. Snakes were captured and officially imported from Kenya. Venom was extracted in the Pata breeding garden (Hlohovec, Slovakia), under the veterinary certificate No. CHEZ-TT-01. This breeding garden also serves as a quarantine station for imported animals and is an official importer of exotic animals from around the world, having the permission of the State Nature Protection of the Slovak Republic under the No. 03418/06, the trade with endangered species of wild fauna and flora and on amendments to certain laws under Law No. 237/2002 Z.z.

In total, 25 mg of crude venom were diluted to the volume of 1 mL with the use of 50 mM sodium acetate pH 5 (buffer A). Ion exchange chromatography was performed on an NGC Chromatography System (Bio-Rad, Hercules, CA, USA) using 6 mL Resource S column (GE Healthcare, Little Chalfont, UK) at 23 °C. For gradient preparation, two buffers were used: A—as described above, and B—50 mM sodium acetate pH 5 with 1M NaCl which served as an eluent. ChromLab software (Bio-Rad, Hercules, CA, USA) was used to monitor the parameters and develop the results.

After optimizing the method, the venom was separated according to the following scheme (Table 4):

Fractions of 1 mL were collected manually and those belonging to one peak were combined and concentrated as described below. To obtain sufficient material for further studies, five identical chromatographic separations were carried out. The initial comparison of the fraction content was made by SDS-PAGE electrophoresis on 13% resolving gels (with 5% stacking gels) according to the standard procedure [82] with a Roti^®^-Mark PRESTAINED molecular weight marker as a standard (Roth, Karlsruhe, Germany) using Mini-Protean II apparatus (Bio-Rad Laboratories, Inc., Hercules, CA, USA). Samples were prepared by mixing 20 µL of pre-concentrated fraction and 10 µL of buffer to SDS-PAGE, and after 5 min of boiling, the samples were put into gel slots. After electrophoresis, the gels were incubated overnight in staining solution with colloidal Coomassie Brilliant Blue G-250. The number of the bands in particular lines was determined using ImageJ software. 

Finally, the peaks from individual chromatographic separations were combined and concentrated on the centrifuge filters Vivaspin 2 with membrane 3000 MWCO PES (Sartorius Stedim Lab Ltd., Stonehouse, UK). The final concentration of the protein in fractions was measured with 2-D Quant Kit (GE Healthcare, Little Chalfont, UK), using bovine serum albumin as a standard.

### 4.2. Sample Preparation for LC-MS/MS

After IEX separation, proteins from each sample (approximately 10) were dissolved in 50 mM ammonium bicarbonate pH 8. Such samples were subjected to acetone precipitation by mixing one volume of samples with six volumes of acetone, and after vortexing, they were incubated overnight at −20 °C. After centrifugation (16,000× *g*, 10 min), the obtained pellets were dissolved with 0.1% (*v*/*v*) RapiGest (Waters, Milford, MA, USA) in 50 mM ammonium bicarbonate pH 8, boiled and then cooled before in-solution digestion. The reduction was carried out with dithiothreitol (DTT) (final concentration: 5 mM) at 60 °C for 30 min, whereas alkylation was performed with iodoacetamide (IAA) (final concentration: 15 mM) for 30 min at room temperature in the dark. The samples were digested with trypsin (50:1 *w*/*w*) (Promega, Madison, WI, USA) for 18 h at 37 °C, and after that, trifluoroacetic acid (TFA) was added to a final concentration of 0.5% (*v*/*v*). Samples were incubated for 45 min at 37 °C and centrifuged for 20 min at 16,000× *g*. Tryptic peptides were then purified on C18 StageTips, which were prepared by packing 6 layers of Empore™ Octadecyl C18 extraction disk (3M, Maplewood, MN, USA) into a 0.2 mL pipette tip. Such columns were successively preconditioned by rinsing with 100% methanol, 60% acetonitrile (ACN) with 0.1% trifluoroacetic acid (TFA), and 0.1% TFA in water. Bound peptides were washed with 5% methanol in 0.1% TFA (1 time) and with 0.1% TFA/H_2_O (3 times) and then eluted with 60% ACN with 0.1% TFA. After every washing and elution step, columns were centrifuged for 5 min at 1000× *g*. Purified samples were evaporated in a vacuum centrifuge, and obtained peptides were dissolved in LC-MS grade water with 0.1% (*v*/*v*) TFA. 

### 4.3. Protein Identification by LC-MS/MS

Approximately 0.6 μg of tryptic peptides from each sample were used for MS analysis. LC separation was conducted on a Dionex Ultimate 3000 Nano system (Thermo Fisher Scientific, Waltham, MA, USA) using Acclaim PepMap RSLC nanoViper C18 column (75 μm × 25 cm, 2 μm particles) (Thermo Fisher Scientific, Waltham, MA, USA) with 180 min ACN gradient (from 4% to 60%, in 0.1% formic acid). Chromatograph worked in on-line mode with Q ExactivePlusOrbitrap mass spectrometer (Thermo Scientific, Waltham, MA, USA). The analysis was conducted in data-dependent acquisition mode with survey scans acquired at a resolution of 70,000 at *m*/*z* 200 in MS mode, and 17,500 at *m*/*z* 200 in MS2 mode. The 15 most prominent peaks from each MS spectra were subjected to further fragmentation. Spectra were recorded in the scanning range of 300–2000 *m*/*z* in positive ion mode. Higher energy collisional dissociation (HCD) ion fragmentation was performed with normalized collision energies set to 25.

Peak lists obtained from MS/MS spectra were identified using 3 search engines: X!Tandem (ver. 2015.12.15.2), MS-GF+ (ver. 2018.04.09), and MyriMatch (ver. 2.2.140). The search was conducted using SearchGUI (ver. 3.3.16) [83]. Protein identification was conducted against a concatenated target/decoy UniProtKB Serpentes database. The decoy sequences were created by reversing the target sequences in SearchGUI. The identification settings were as follows: Trypsin, Specific, with a maximum of 2 missed cleavages; 10.0 ppm as MS1 and 0.02 Da as MS2 tolerances; fixed modifications: Carbamidomethylation of C, variable modifications: Oxidation of M, fixed modifications during refinement procedure: Carbamidomethylation of C, variable modifications during refinement procedure: Acetylation of protein *N*-term, Pyrolidone from E, Pyrolidone from Q, Pyrolidone from carbamidomethylated C.

Peptides and proteins were inferred from the spectrum identification results using PeptideShaker (ver. 1.16.42). Peptide spectrum matches (PSMs), peptides, and proteins were validated at a 2.5% false discovery rate (FDR) estimated using the decoy hit distribution. Hits marked by the software as “validated” were taken into consideration in further analysis, whereas, proteins labeled as “doubtful” were manually revised and based on the #PSMs, #Peptides and the obtained spectra, some of them were also included into data analysis.

Quantification of proteins was performed using the NSAF+ algorithm, which was implemented into PeptideShaker software [84]. Data was extracted into Excel and then the identified proteins were assigned to different groups/families. Final quantitative values for the whole protein group/family were calculated by summing individual NSAF+ values of proteins assigned to a given group. 

The mass spectrometry data along with the identification results have been deposited to the ProteomeXchange Consortium via the PRIDE partner repository with the dataset identifier PXD015814 and 10.6019/PXD015814.

### 4.4. Antibacterial Activity

The following certified bacteria cultures used to test the antimicrobial activity were received from the Department of Biotechnology and Bioinformatics, Faculty of Chemistry, Rzeszow University of Technology (*S. epidermidis* ATTCC 12228, does not form a biofilm) and from Chair and Department of Medical Microbiology Medical University of Lublin (*S. epidermidis* ATCC 35984, form a biofilm). The anti-biofilm activity of tested fractions was evaluated with the use of three clinical Methicyllin-Resistant Coagulase-Negative Staphylococci (MRCNS) strains of (2346, 2452, 2702) obtained from the Department of Medical Laboratory Diagnostics of Provincial Specialist Hospital in Rzeszow. 

#### 4.4.1. Determination of Minimum Inhibitory Concentration (MIC) 

The antibacterial activity of fractions (1–10) was evaluated by determination of the minimum inhibitory concentration (MIC, μg/mL) using the micro-broth dilution method, as described before [85] and according to the Clinical and Laboratory Standards Institute (CSLI) guidelines for *S. epidermidis* antimicrobial susceptibility testing [86]. Briefly, each bacterial strain was incubated in 37 °C in New Brunswick Innova 40 Shaker (Eppendorf AG, Hamburg, Germany) until turbidity of 0.5 McFarland’s standard (10^8^ CFU/mL, colony-forming units per mL) was obtained. Series of two-fold dilutions of tested fractions (range from 19 to 500 μg/mL) were prepared in Muller Hinton Broth (MHB). Working bacterial cultures were diluted to final density 10^5^ CFU/mL, added to prepared series of fraction’s dilutions, and incubated at 37 °C. After 24 h, bacterial growth in comparison with the positive control (the medium without antibacterial agents) was monitored. The MIC was defined as the lowest concentration of the antibacterial agent, which completely inhibited the visible growth of the microorganism. These results were confirmed by measurement of the optical density at 630 nm using BIO-RAD Microplate Reader. The experiment was carried out in triplicate. A positive (the medium without antibacterial agents) and negative control (no bacterial cultures added) of bacterial growth and solvent control were performed. An evaluation of the antibiotic susceptibility of each bacterial strain to ampicillin (AMP), tetracycline (TET), kanamycin (KAN), streptomycin (STR), and chloramphenicol (CF) was also performed by the micro-dilution method (range from 0.03 to 500 μg/mL) All reagents and bacterial cultures were prepared using Laminar Flow Cabinet ESCO Airstream.

#### 4.4.2. Synergy Testing 

Two standard *S. epidermidis* (ATCC12228 and ATCC35984) isolates were used to test interactions between two antibacterial agents (antibiotic and fraction F2) by the checkerboard assay [87,88] using the same medium and incubation conditions as described for MIC determinations. Two out of five antibiotics (ampicillin, tetracycline) were selected to the checkerboard assay due to: (i) Limited amount of fraction F2; (ii) the lowest (tetracycline) and the highest (ampicillin) obtained MIC values; (iii) different mechanisms of action of ampicillin (as beta lactam antibiotic) and tetracycline (as tetracycline antibiotic); and (iv) different activity of ampicillin (bactericidal) and tetracycline (bacteriostatic) [46]. Kanamycin and streptomycin were excluded due to resistance to at least one certified *S. epidermidis* strain. Compounds were usually tested in a range from 1/32 to 4 × MIC. The checkerboard assay was conducted on 96-well microtiter plates, where the first antimicrobial agent (fraction F2) in the combination was serially diluted 2-fold along the abscissa, whereas the second (antibiotic) was diluted along the ordinate. Starting concentrations for 2-fold dilution of compounds were selected on the basis of previously determined MIC. The initial concentration of fraction F2 in combination with both antibiotics was 37.25 and 74.5 µg/mL, respectively, against *S. epidermidis* 12228 and *S. epidermidis* 12228. Against *S. epidermidis* 12228, starting concentrations of ampicillin and tetracycline in combination with fraction 2 were 250 µg/mL. Ampicillin and tetracycline were used with an initial concentration of 500 and 0.49 µg/mL, respectively, against *S. epidermidis* 35984. The MIC of every antimicrobial agent in combination represented the lowest dilution that completely inhibited the growth of the bacterium. The interaction of the drugs in a combination was expressed quantitatively as a fractional inhibitory concentration (FIC) Index (FICI) and calculated for each drug combination using the following equation: FICI = FICA + FICB, where FICA = MIC of drug A in the combination/MIC of drug A alone, and FICB = MIC of drug B in the combination/ MIC of drug B alone. The FICI results were interpreted as synergistic (≤0.5), additive (>0.5 to ≤1), neutral (1–2), or antagonistic (≥2) [89]. 

#### 4.4.3. Anti-Biofilm Activity

*Staphylococcus epidermidis* certified and three clinical strains were cultured overnight at 37 °C in MHB. Series of two-fold dilutions in MHB of fraction F2 and proper bacterial culture (10^5^ CFU/mL) were added to 96-well polystyrene microtiter plates and incubated in 37 °C. The final concentrations of the tested compounds ranged from 0.58 to 149 μg/mL. The negative control was MHB medium, and the positive control (biofilm formation) was bacterial culture in MHB. After incubation, medium was removed from wells and washed twice with sterile phosphate-buffered saline (PBS) to remove the planktonic bacteria. Alive and adherent bacterial cells that usually formed biofilm in each well of the microtiter plate were stained with 3-(4,5-dimethyl-2-thiazolyl)-2,5 diphenyl-2*H*-tetrazolium bromide (MTT; 0.5% in PBS) for 2 h at 37 °C (protected from light) [90]. After incubation, the solution was removed, and bacterial biofilm was solubilized by DMSO and mixed for 15 min at room temperature in an INNOVA 40 Incubator Shaker. The absorbance was measured at 630 nm using a spectrophotometer (BIO-RAD Microplate Reader). No biofilm-producing *S. epidermidis* ATCC 12228 and high biofilm-producing *S. epidermidis* ATCC 35984 were used, respectively, as a negative and positive control. Tetracycline was used as the reference antimicrobial compound with the final concentration ranging from 0.03 to 500 μg/mL. The amount of biofilm inhibition was calculated relative to the amount of biofilm grown in the absence of anti-biofilm agent (defined as 100%) and the media sterility control (defined as 0%). Results from at least three separate biological replicates were averaged.

#### 4.4.4. Statistical Analysis

The final results obtained from triplicate experiments were presented as means ± SD. STATISTICA v.12 (StatSoft, OK, USA) software was used to analyze the significances between data. *p* values were calculated using one-way ANOVA to compare the differences between each pair of groups. *p* value <0.05 was considered as significant.

## 5. Conclusions

Fractions of *Naja ashei* venom differing in protein composition were tested against certified and clinical strains of *Staphylococcus epidermidis*. Fraction F2 composed mostly with PLA_2_s, 3FTxs, CRISPs, LAAOs, alkaline phosphatase-like proteins, and Ig-like domain-containing proteins was the only one with antibacterial properties. It also showed a synergistic effect with antibiotics and was effective in the inhibition of biofilm formation. Among the F2 components, PLA_2_s, 3FTxs, and LAAOs were described before as potential components with bacteriostatic and bacteriocidal properties. The main thing that distinguishes this fraction from the others is the high content of L-amino acid oxidases and it is in this fact that the unique properties of this fraction are probably to be seen. Moreover, after separating *Naja ashei* venom into fractions, low-copy proteins with the Ig-like domain (SSF48726) were identified. The proteins belonging to this superfamily are characterized by a great variety of functions and sequences, but they are united by the type of folding. Most of the identification was made based on the transcript sequences from the venom glands, which suggests that these proteins are not related to the immune system but their function remains unexplained.

## Figures and Tables

**Figure 1 molecules-25-00293-f001:**
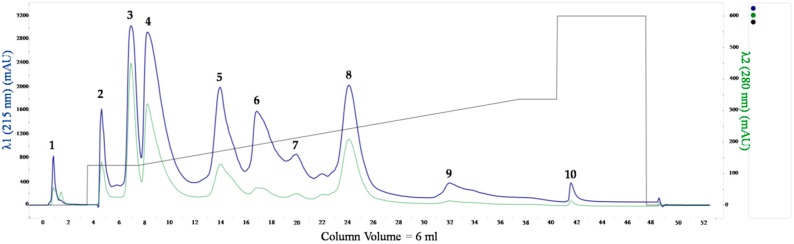
Representative chromatogram obtained for the separation of *Naja ashei* venom on the Resource S column. The fractions considered in the next stages of the experiment are marked by numbers above the peaks.

**Figure 2 molecules-25-00293-f002:**
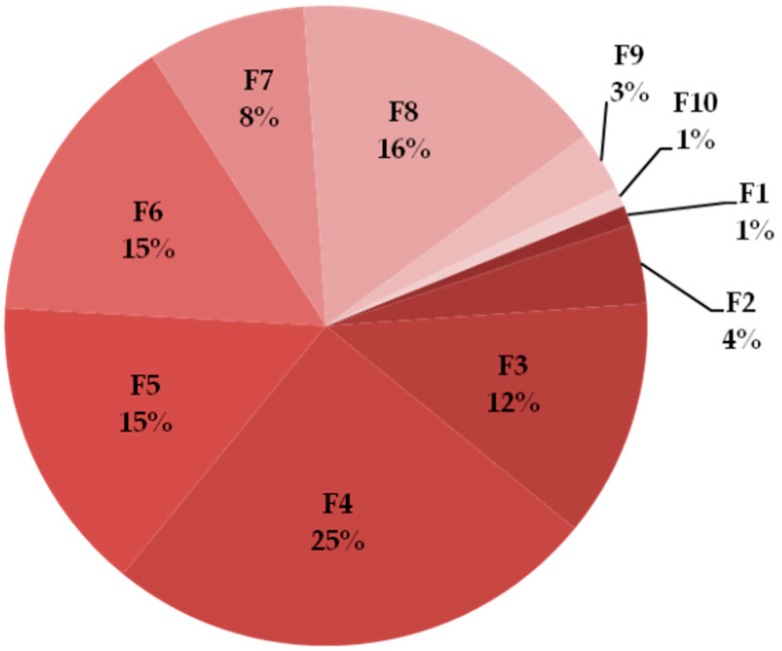
Percentage share of collected proteins in particular fractions from AUC analysis in ChromeLab.

**Figure 3 molecules-25-00293-f003:**
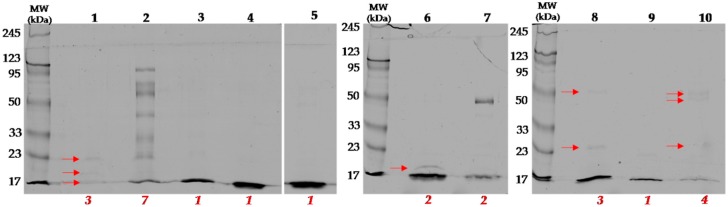
Representative SDS-PAGE gels of *Naja ashei* venom fractions. The numbers above the lines represent the fractions collected. The numbers below lines (red) indicate the number of bands defined in the ImageJ software. Weakly visible bands are marked with arrows on the gels.

**Figure 4 molecules-25-00293-f004:**
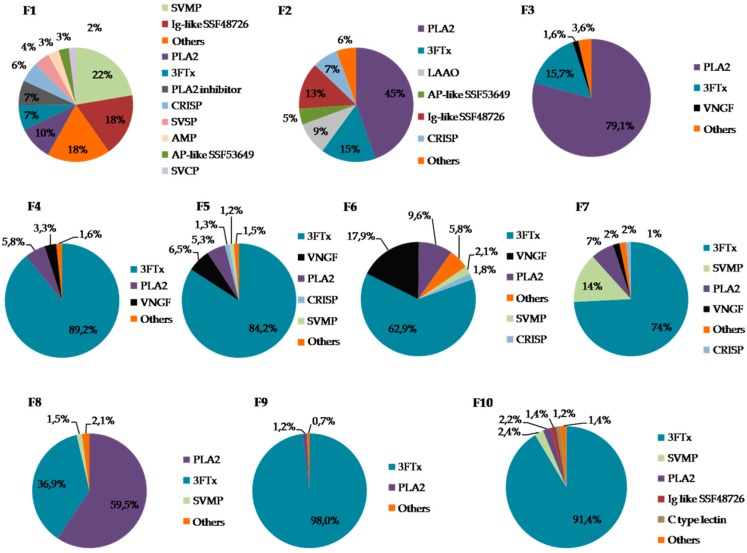
The percentage distribution of particular protein groups in each fraction estimated using semi-quantitative MS analysis. (SVMP—Snake venom metalloproteinase, Ig-like SSF48726—Immunoglobulin-like domain-containing protein; PLA2—Phospholipase A_2_, 3FTx—Three-finger toxin, CRISP—Cysteine-rich secretory protein, SVSP—Snake venom serine proteases, AMP—Antimicrobial peptide, AP-like SSF53649—Alkaline phosphatase-like protein, SVCP—Snake venom cysteine protease, LAAO—L-amino acid oxidase, VNGF—Venom nerve growth factor).

**Figure 5 molecules-25-00293-f005:**
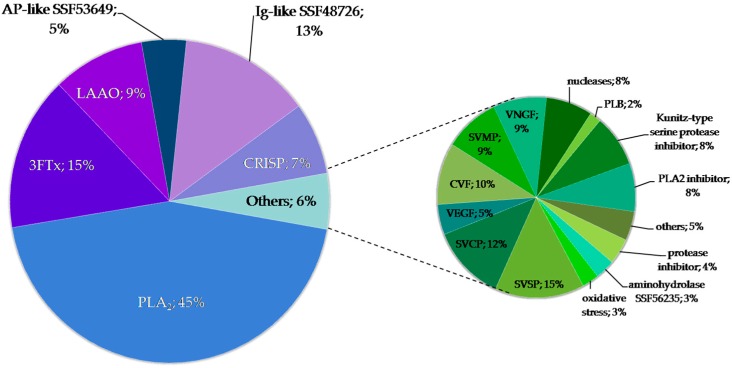
Detailed percentage distribution of identified proteins in fraction F2. LAAO—L-amino acid oxidase, 3FTx—3 finger toxin, PLA_2_—phospholipase A_2_, AP—Alkaline phosphatase -like SSF53649, CRISP—cysteine-rich secretory protein, SVMP—snake venom metaloproteinase, SVSP—snake venom serine proteinase, SVCP—snake venom cysteine proteinase, CVF—cobra venom factor, VNGF—venom nerve growth factor, VEGF—vascular endothelial growth factor.

**Figure 6 molecules-25-00293-f006:**
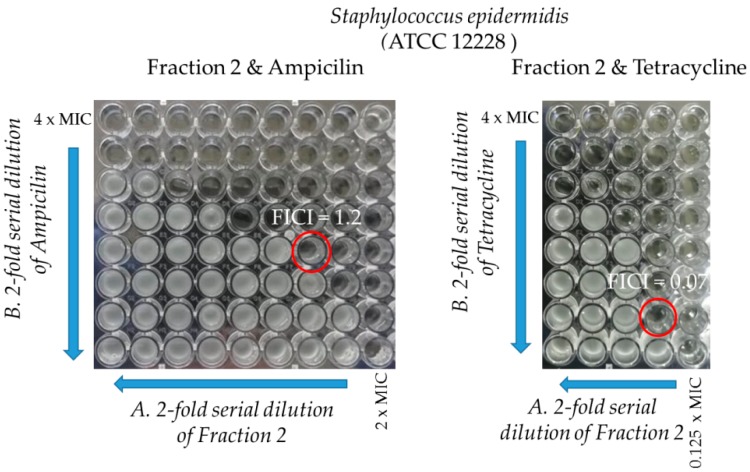
Effect of the combination of fraction F2 with antibiotics: ampicillin (left) and tetracycline (right) against certified *S. epidermidis* ATCC 12228. FICI arrows indicate the direction of the decreasing concentration. Fraction F2 MIC = 37.25 µg/mL.

**Figure 7 molecules-25-00293-f007:**
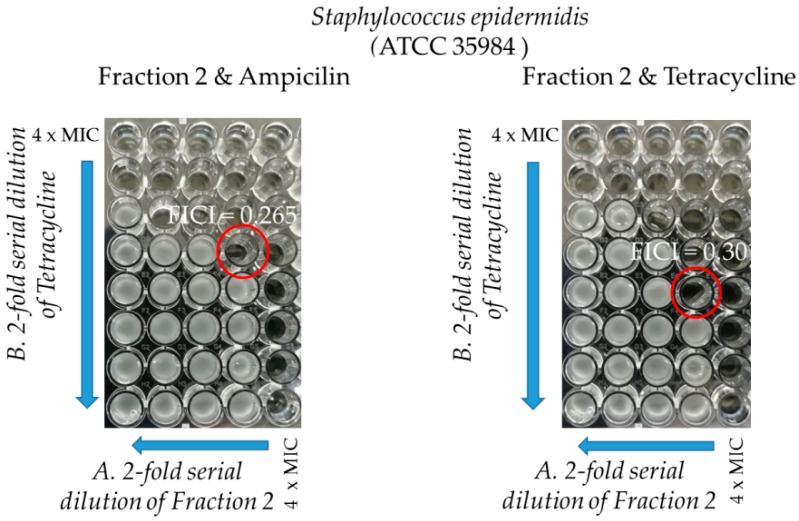
Effect of the combination of fraction F2 with antibiotics: ampicillin (left) and tetracycline (right) against certified *S. epidermidis* ATCC 35984. FICI arrows indicate the direction of the decreasing concentration. Fraction F2 MIC = 9.3 µg/mL.

**Figure 8 molecules-25-00293-f008:**
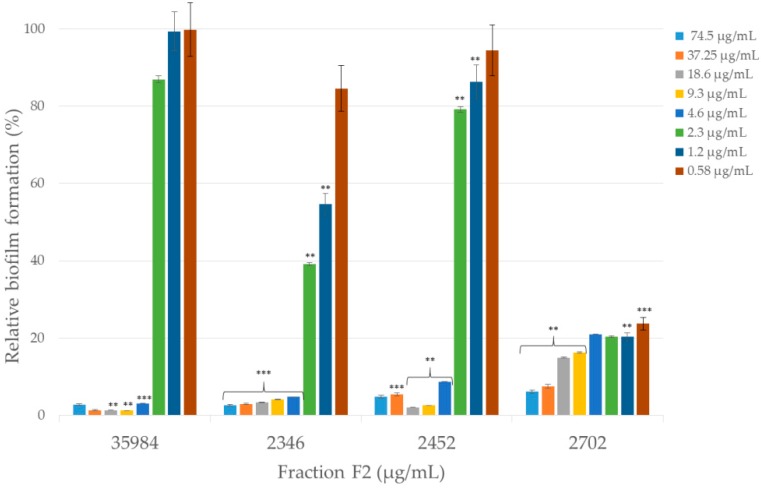
Anti-biofilm activities of fraction F2 against standard *S. epidermidis* (ATCC 35984) and clinical strains (2346, 2452, 2702). Fraction F2 occurred in two-fold decreasing concentrations (74.5, 37.25, 18.6, 9.3, 4.6, 2.3, 1.2, and 0.58 µg/mL) for each bacterial strain. The error bars and asterisks represent the standard errors and statistical significance with the *p*-values: *** *p* < 0.001, ** *p*< 0.01.

**Table 1 molecules-25-00293-t001:** Low abundant proteins in *Naja ashei* venom. (+) denotes fractions in which a given protein group was detected; (−) denotes fractions in which the presence of a given protein group was not observed.

	Fraction									
Protein	1	2	3	4	5	6	7	8	9	10
C-type lectin	−	−	−	−	−	−	−	−	−	+
SVSP	+	+	−	−	−	−	−	−	−	+
SVCP	+	+	+	−	−	−	−	−	−	−
PLA_2_ inhibitor	+	+	−	−	−	−	−	−	−	+
AP-like SSF53649	+	+	−	−	−	−	−	−	−	−
LAAO	−	+	+	−	−	−	−	−	−	+
Ig-like SSF48726	+	+	+	−	−	−	−	−	−	+
CRISP	+	+	+	+	+	+	+	+	−	−
VNGF	+	+	+	+	+	+	+	+	+	+
AMP	+	−	−	−	−	−	−	−	−	−
CVF	+	+	+	+	+	+	+	+	−	+

**Table 2 molecules-25-00293-t002:** MIC of antibiotics and fraction F2 of *Naja ashei* venom on *S. epidermidis* certified and clinical strains.

Antibiotic	*Staphylococcus epidermidis*				
	ATCC 12228	ATCC 35984	2346	2452	2702
	MIC µg/mL				
Chloramphenicol	7.8	15.6	125	7.8	125
Ampicillin	62.5	250	500	62.5	7.8
Kanamycin	1.9	−^a^	−^a^	1.9	3.9
Streptomycin	−^a^	−^a^	3.9	7.8	250
Tetracycline	62.5	0.12	1.9	62.5	0.24
Fraction F2	37.25	9.3	4.6	9.3	37.25

^a^—no inhibition of growth in the concentration range.

**Table 3 molecules-25-00293-t003:** MICs and FIC indexes of fraction F2 with antibiotics against *S. epidermidis* standard strains.

Strains	Ampicillin/FractionF2			Tetracycline/FractionF2		
	MIC (µg/mL)					
	In Single Use	In Combination	FIC Index	In Single Use	In Combination	FIC Index
ATCC 12228	62.5/37.25	15.6/37.25	1.2	62.5/37.25	3.9/2.3	0.07
ATCC 35984	250/9.3	62.5/2.3	0.26	0.12/9.3	0.03/2.3	0.30

**Table 4 molecules-25-00293-t004:** *N. ashei* venom fractionation program.

Step	Segment	Initial %B	Final %B	FR (mL/min)	CV
Equilibration	Isocratic	0	0	6	3
Sample application	Isocratic	0	0	3	0.5
Column wash	Isocratic	0	0	6	3
Elution	Isocratic	21	21	6	4
	Gradient	21	56	6	30
	Isocratic	56	56	6	3
	Isocratic	100	100	6	7
Column wash	Isocratic	0	0	6	5

B—buffer B (50 mM sodium acetate pH 5 with 1M NaCl), FR—flow rate, CV—column volume.

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
