# Peer review of "Antimicrobial Activity of Protein Fraction from Naja ashei Venom against Staphylococcus epidermidis"

_molecules, 2020, doi:10.3390/molecules25020293_

Round 1

Reviewer 1 Report

The research article “Antimicrobial activity of protein fraction from Naja ashei venom against Staphylococcus epidermidis by Bocian etal highlights the need of antimicrobial substances to combat microbial infections.  This is an interesting research work is interesting.

1). Authors have missed the relevant literature on snake venom peptides as antimicrobial agents. It will be important for the readership to include the following literature

https://www.ncbi.nlm.nih.gov/pubmed/26930579

https://www.ncbi.nlm.nih.gov/pubmed/30053390

https://www.ncbi.nlm.nih.gov/pubmed/29782980

in the introduction and in discussion of manuscript.

2). Some information on the effectiveness and importance of venom peptides in therapeutic use and possible molecular targets should be introduced and discussed.

3) What is the possible molecular mechanism of F2 fraction in the present study? The discussion part misses the possible molecular target (s), honeybee venom and snake venoms are documented to inhibit bacterial ATP synthase. At least authors should acknowledge that.

4) What is the purity and ratio of F2 fraction in synergistic experiments?

Author Response

The research article “Antimicrobial activity of protein fraction from Naja ashei venom against Staphylococcus epidermidis by Bocian etal highlights the need of antimicrobial substances to combat microbial infections.  This is an interesting research work is interesting.

1). Authors have missed the relevant literature on snake venom peptides as antimicrobial agents. It will be important for the readership to include the following literature

https://www.ncbi.nlm.nih.gov/pubmed/26930579

https://www.ncbi.nlm.nih.gov/pubmed/30053390

https://www.ncbi.nlm.nih.gov/pubmed/29782980

in the introduction and in discussion of manuscript.

Information about AMP was included in the manuscript.

2). Some information on the effectiveness and importance of venom peptides in therapeutic use and possible molecular targets should be introduced and discussed.

We agree that AMPs have great value as molecular targets and are important antimicrobial agents, but we did not detect the presence of those peptides in fraction F2. That is why in our opinion details about this group of venom components should not be discussed in our manuscript.

3) What is the possible molecular mechanism of F2 fraction in the present study? The discussion part misses the possible molecular target (s), honeybee venom and snake venoms are documented to inhibit bacterial ATP synthase. At least authors should acknowledge that.

Discussion section was supplied with speculation about potential F2’s mechanism of action. None of the F2 fractional components have ATP synthase inhibitor activity so this topic has not been addressed.

4) What is the purity and ratio of F2 fraction in synergistic experiments?

Composition of the fraction F2 is presented on the Figure 4 and 5. Information about ratio of F2 and antibiotics was detailed in section Materials and Methods (4.4.2)

Reviewer 2 Report

The quality of figure 1 could be improved, axis not readable.

The colour scheme for the fraction in figure 4 is not consistent (CRISP have different colors).

Define SVMP other abbreviation when used for the first time (line 107)

Supplementary information only has S3, Please provide S1 and S2

The authors do not mention how was the concertation of the F2 was determined. Was it by mass or by measuring A280?

Why was tetracycline and ampicillin selected for the synergy testing? The mechanism of action of these antibiotics should be briefly mentioned in few sentences in the discussion.  

An Explanation/discusiion for F1 that had AMPs but did not show antimicrobial activity?

Conclusion mentions biofilm eradication (line 472) but the assay performed was biofilm inhibition, Can the authors perform a biofilm eradication assay as described in kumar et.al infectious disease, 2018

Interestingly F2 fraction has a high abundance of few proteins (figure 3 and 4) such CRISP and LAAO, but Fraction 1 has a low abundance of these proteins. The authors attribute the antimicrobial activity of F2 to LAAO alone; however, synergistic effect should be also considered and discussed.

A possible experiment would take the F2 fraction, further purify, and isolate the active ingredient so that the antimicrobial activity can be attributed to LAAO.

Overall, the paper is well written and results are discussed appropriately. 

Author Response

The quality of figure 1 could be improved, axis not readable.

Figure F1 was corrected.

The colour scheme for the fraction in figure 4 is not consistent (CRISP have different colors).

We apologize for this mistake. Wrong version of the figure was included in the final version of the manuscript.

Define SVMP other abbreviation when used for the first time (line 107).

Done

Supplementary information only has S3, Please provide S1 and S2

Our zip file contains all three files. We cannot explain why only S3 is visible. We prepared zip again and upload it once again.

The authors do not mention how was the concertation of the F2 was determined. Was it by mass or by measuring A280?

Proper description was added in the Methods section. Information about protein concentration was also added in paragraph 2.2.2.

Why was tetracycline and ampicillin selected for the synergy testing? The mechanism of action of these antibiotics should be briefly mentioned in few sentences in the discussion.  

All information was added to the methodology and discussion sections.

An Explanation/discusiion for F1 that had AMPs but did not show antimicrobial activity?

The subject was additionally discussed.

Conclusion mentions biofilm eradication (line 472) but the assay performed was biofilm inhibition, Can the authors perform a biofilm eradication assay as described in kumar et.al infectious disease, 2018

As the Reviewer 2 rightly noted, we performed biofilm inhibition instead of biofilm eradication assay, and the necessary correction was therefore made in the text.

At the stage of methodology development we planned to conduct a biofilm eradication assay [Vishwakarma and Vavilala, 2019]. However, we decided that it is more important to acquire knowledge about the anti-biofilm potential properties of peptides to prevent biofilm-related infection associated with various types of medical devices (e.g. catheters, surgical vascular grafts, joint prostheses, heart valves). The methodology was based on Bielenica et al. (2018). Practically we cannot perform the biofilm eradication assay due to completion of the planned experiments and a lack of faction F2.

Interestingly F2 fraction has a high abundance of few proteins (figure 3 and 4) such CRISP and LAAO, but Fraction 1 has a low abundance of these proteins. The authors attribute the antimicrobial activity of F2 to LAAO alone; however, synergistic effect should be also considered and discussed.

The subject was additionally discussed.

A possible experiment would take the F2 fraction, further purify, and isolate the active ingredient so that the antimicrobial activity can be attributed to LAAO.

Yes, we definitely agree. This is the purpose of our future research. In our plans is cloning of particular components and tests of their properties alone and in combinations.

Overall, the paper is well written and results are discussed appropriately. 

Thank you very much for the kind words. We are sure that thanks to your comments we have managed to improve our article.

Round 2

Reviewer 1 Report

Accept the revised version